Applied and Environmental Science
# Phylogenetic Distribution of Plastic-Degrading Microorganisms

Victor Gambarini,[a] Olga Pantos,[b] Joanne M. Kingsbury,[b] Louise Weaver,[b] Kim M. Handley,[a] Gavin Lear[a]

aSchool of Biological Sciences, University of Auckland, Auckland, New Zealand
bThe Institute of Environmental Science and Research, Ilam, Christchurch, New Zealand

**ABSTRACT** The number of plastic-degrading microorganisms reported is rapidly increasing, making it possible to explore the conservation and distribution of presumed plastic-degrading traits across the diverse microbial tree of life. Putative degraders of conventional high-molecular-weight polymers, including polyamide, polystyrene, polyvinylchloride, and polypropylene, are spread widely across bacterial and fungal branches of the tree of life, although evidence for plastic degradation by a majority of these taxa appears limited. In contrast, we found strong degradation evidence for the synthetic polymer polylactic acid (PLA), and the microbial species related to its degradation are phylogenetically conserved among the bacterial family *Pseudonocardiaceae*. We collated data on genes and enzymes related to the degradation of all types of plastic to identify 16,170 putative plastic degradation orthologs by mining publicly available microbial genomes. The plastic with the largest number of putative orthologs, 10,969, was the natural polymer polyhydroxybutyrate (PHB), followed by the synthetic polymers polyethylene terephthalate (PET) and polycaprolactone (PCL), with 8,233 and 6,809 orthologs, respectively. These orthologous genes were discovered in the genomes of 6,000 microbial species, and most of them are as yet not identified as plastic degraders. Furthermore, all these species belong to 12 different microbial phyla, of which just 7 phyla have reported degraders to date. We have centralized information on reported plastic-degrading microorganisms within an interactive and updatable phylogenetic tree and database to confirm the global and phylogenetic diversity of putative plastic-degrading taxa and provide new insights into the evolution of microbial plastic-degrading capabilities and avenues for future discovery.

**IMPORTANCE** We have collated the most complete database of microorganisms identified as being capable of degrading plastics to date. These data allow us to explore the phylogenetic distribution of these organisms and their enzymes, showing that traits for plastic degradation are predominantly not phylogenetically conserved. We found 16,170 putative plastic degradation orthologs in the genomes of 12 different phyla, which suggests a vast potential for the exploration of these traits in other taxa. Besides making the database available to the scientific community, we also created an interactive phylogenetic tree that can display all of the collated information, facilitating visualization and exploration of the data. Both the database and the tree are regularly updated to keep up with new scientific reports. We expect that our work will contribute to the field by increasing the understanding of the genetic diversity and evolution of microbial plastic-degrading traits.

**KEYWORDS** biodegradation, phylogenetic distribution, plastic

Since the development of the first synthetic polymer in 1907 (1), plastics have become indispensable to humanity. Even though the mass production of plastics dates back only to the early 1940s, the extraordinary and rapid growth in their

Address correspondence to Gavin Lear, g.lear@auckland.ac.nz.

We have centralized information on known plastic degrading microorganisms within an interactive and updatable phylogenetic tree and database, which shows the global and phylogenetic diversity of plastic-degrading taxa and enzymes.

production is almost unparalleled among man-made materials (2). Global plastic production reached 368 million tons per annum in 2019, a figure which is predicted to double over the next 20 years (3–5). Through atmospheric and oceanic transport, plastics have become globally ubiquitous and can be found in abundance from the deepest marine ecosystems on earth (6) to remote and pristine mountains, where they are deposited as fine "snow" (7, 8). Plastic waste is now so abundant in our environment that it is considered a defining characteristic of our Anthropocene era (9).

Largely driven by the disposal of single-use plastics, the relative mass of plastics in municipal solid waste has increased from less than 1% in the 1960s to more than 10% in the 2000s (10). The most produced plastic types are polyethylene (PE) (36%), polypropylene (PP) (21%), and polyvinylchloride (PVC) (12%), in addition to polyethylene terephthalate (PET), polyurethane (PU), and polystyrene (PS), with <10% each (2). Improper disposal is causing plastic waste to accumulate in terrestrial and marine environments (11). For example, it has been estimated that between 5 million and 13 million tons of plastic litter enter the world's oceans annually from coastal countries, with the top 20 polluting rivers contributing to approximately two-thirds of river emissions (12–14). The impacts of synthetic plastic polymers on aquatic life have already been reported for over 700 species, ranging from microscopic phytoplankton at the base of the food web (15) to whales (16). Predominant impacts relate to plastic ingestion, decreased nutrition from intestinal blockage, and decreased mobility (5); animals with the most reports of entanglement or ingestion are sea turtles, marine mammals, and seabirds (17). Plastic polymers and the additives frequently blended into commercial-grade plastics have also been shown to accumulate in marine species harvested for human consumption (17–20). Nearly one-fifth of all marine life with records of plastic ingestion or entanglement are threatened or near threatened according to the International Union for Conservation of Nature (IUCN) list of threatened species (21, 22). Small plastic particles even accumulate around seed pores, delaying the germination and growth of terrestrial vascular plants (23). Although the impacts of plastics on macroorganisms are increasingly documented in the literature, little is known regarding how microorganisms interact with synthetic polymers in the environment.

Plastic waste impacts the composition and activity of microbial communities and taxa (24). To date, only a small number of microorganisms have been shown to exhibit the ability to degrade plastic polymers, the first being reported roughly 30 years after mass production began (25). Since then, the number of known plastic-degrading microorganisms has continued to increase and includes notable bacteria such as *Ideonella sakaiensis* (26). *I. sakaiensis* produces a newly identified polyethylene terephthalatase (PETase) enzyme with specificity toward PET degradation, as detailed in a recent review by Salvador et al. (27). Examples of fungal degraders of plastic include *Parengyodontium album*, reported to degrade polymers of polylactic acid (28). Among major synthetic polymers, PET is the most documented polymer, with a recently published study shedding light on the phylogenetic relationships, the recent evolution, and the global distribution of PET hydrolases (29). Despite some attempts to summarize and consolidate knowledge concerning the diversity and distribution of plastic-degrading microorganisms (see references 30–36), the phylogenetic distribution of organisms capable of degrading the multitude of plastic polymers in current circulation remains little explored. Taxonomic clades of organisms with an increased likelihood of possessing plastic-degrading traits have not been identified, and the degradative mechanisms utilized by many plastic-degrading taxa remain uncharacterized, hampering the discovery of novel plastic-degrading enzymes.

To uncover and organize current knowledge regarding the microbial degradation of plastics, we conducted a comprehensive literature review and compiled information on the microbial species, types of plastic degraded, and respective references in a readily updatable and interactive phylogenetic tree. Using these data, we assessed if the ability of microorganisms reported to degrade different plastics is phylogenetically conserved or widely distributed among phylogenetically disparate taxa. Additionally,

we mined available genomes of microbial taxa with reported plastic-degrading capabilities to identify the presence of candidate plastic-degrading genes and highlight novel avenues for bioprospecting plastic-degrading microbial enzymes.

## RESULTS AND DISCUSSION

A total of 1,451 publications mentioning microbial plastic degradation were identified (see Materials and Methods for literature search terms). On further inspection, 408 of these articles were verified as describing plastic-degrading microbes (see Materials and Methods for literature search terms and requirements for studies to be deemed sufficient to demonstrate putative plastic degradation). Up to April 2020, the total number of species reported to have plastic-degrading capabilities using our search terms was 436, the first being described in a publication in 1974 (37). In 37 years of research, from 1974 to 2010, 219 species of microorganisms were reported to degrade plastics. This number almost doubled in the following 10 years, reaching 436 species reported up to April 2020 (see Fig. S1 in the supplemental material). Of the 66 different types of plastics evaluated in the literature, the species reported to degrade the most types were *Bacillus pumilus*, *Aspergillus fumigatus*, and *Phanerochaete chrysosporium*, each having been shown to degrade 14, 11, and 10 different types of plastic, respectively. It is important to note that some plastics contain additives that may inhibit microbial growth (38) such that the number of degraders of pure plastic polymers could currently be underestimated. Conversely, numerous studies may mistakenly report the microbial degradation of plastic additives or only low-molecular-weight polymers of monomers of various plastics, as recently described by Danso et al. (33). Most putative plastic degraders were isolated from soils (27.8%), plastic waste dumping sites (9.6%), and composts (5.3%), while a considerable proportion was obtained from culture collections of microorganisms (15.9%) (Fig. S2A). The countries that reported the most isolation of plastic degraders were Japan (14.1%) and India (13.8%) (Fig. S2B).

We created an interactive phylogenetic tree to visualize the relationship of suggested degraders of each of 66 different plastic types assessed by the study. By plotting each taxon's reported ability to degrade various plastics, we are able to demonstrate that plastic-degrading traits are widely dispersed across the microbial tree of life (Fig. 1). Among the most commonly produced synthetic polymers, polyethylene (PE) was reported as being degraded by the most plastic-degrading taxa known to date, including 55 bacterial and 24 fungal species. Despite the dominance of polypropylene (PP) and polystyrene (PS) in global plastic production, few organisms as yet appear to have been identified with the ability to degrade their high-molecular-weight polymers. PP was reported to be degraded by 2 fungal and 12 bacterial species, of which 8 species belonged to the *Bacillales*. The distribution was similar for PS, which is suggested to be degraded by at least 1 fungal and 14 bacterial species. Among the 14 bacterial species, 7 were similarly identified as belonging to the *Bacillales*, highlighting this order as a possible source of additional polypropylene- and polystyrene-degrading bacteria. These PP- and PS-degrading *Bacillales* were isolated from diverse environments. The 14 PP degraders were described by six different reports and were isolated from soil (39, 40), mangrove sediments (41, 42), sewage treatment plants, and municipal landfills (43). Furthermore, the 15 putative PS degraders were described by seven different studies and were isolated from mangrove sediments (41), soil (44), mealworm guts (45), wetland water (46), and a plastic waste yard (47). Reports of the degradation of PS and PP must be treated with caution, however, since evidence for the degradation of higher-molecular-weight polymers remains limited, as it does for other polyamide and PVC polymers (33). In most studies, substantial further evidence is required to confirm the degradation of the polymer rather than residual biodegradable monomers such as styrene (48) or plastic additives, which comprise a substantial fraction of some plastics, particularly PVC.

We confirm the natural polymer polyhydroxybutyrate (PHB) as being degraded by the greatest number of species: 126 different bacterial and fungal species were

suggested to degrade this natural polymer. At least 94 microorganisms are similarly identified as being capable of degrading the synthetic polymer polylactic acid (PLA) (31 fungal and 63 bacterial species), possibly reflecting the fact that the stereochemical positions of the chiral carbons of the L-lactic acid unit of PLA and the L-alanine unit in the silk fibroin are similar (49). Silk fibroin, a fibrous protein produced by domestic silkworms, is rich in glycine, alanine, and serine (50) and is a natural analog of poly(L-lactide). In fact, the bacterial genus *Amycolatopsis*, which had the largest number of PLA degraders reported for a single genus in our data set, also has several species reported to degrade both PLA and silk fibroin (51–53). Hence, microorganisms that are able to degrade PLA likely identify the L-alanine unit in silk fibroin as an analog of the L-lactate unit in PLA (49). PLA is widely marketed as a biodegradable plastic with the potential to be transformed by enzymatic microbial activity, under aerobic conditions, into water and carbon dioxide. Many bacteria with the capacity to degrade PLA belong to the *Pseudonocardiaceae* (Fig. 1) (54, 55). Twenty-five PLA-degrading *Pseudonocardiaceae* species are reported in 11 different publications, all of which were either isolated and tested directly from soil (56, 57) or obtained from bacterial culture collections (53). Those species from culture collection isolates had originally been isolated from soils (55, 58) as well as from diverse environments such as the rumen of cattle (51, 54).

To verify whether plastic degradation traits are clustered within certain taxonomic groups or spread around the tree of life, we created a phylogenetic tree using 7,000 bacterial and fungal species randomly sampled from the full National Center for Biotechnology Information (NCBI) taxonomy database. We also included the 436 reported plastic degraders identified by our analyses (Fig. S3). Overall, plastic degradation traits appeared to be dispersed across the microbial tree of life; most major bacterial and fungal phyla (i.e., those containing the highest number of known taxa) have species reported to degrade plastic. However, no representatives of *Archaea* are reported to have plastic-degrading capabilities, nor are many smaller bacterial and fungal phyla (e.g., *Acidobacteria* and *Spirochaetes*) (Fig. S4). The lack of reports of plastic degradation by species from the domain *Archaea* and several bacterial and fungal phyla may be due to these groups containing fewer documented taxa overall. Alternatively, these species may be more challenging to grow under laboratory conditions. For instance, the phylum "*Candidatus* Saccharibacteria" has been found repeatedly in many different environments, but significant cultivation attempts have so far been unsuccessful (59). Methods to isolate plastic degraders likely favor the isolation of certain bacterial and fungal strains; hence, the plastic-degrading potential of other taxonomic groups may be yet to be discovered.

The relationship between phylogeny and the degradation of each type of plastic was examined to determine if plastic degradation traits are randomly distributed or phylogenetically constrained among known degraders. The statistic "$D$" test was utilized to measure the phylogenetic signal in binary traits, as proposed by Fritz and Purvis (60) and implemented in the R package caper (61). Here, the binary traits tested were the presence or absence of degradation reports for each taxon included in the tree for each type of plastic. The strongest phylogenetic signal of the plastic types was for PLA degradation ($D = 0.54$ and $P_{Brownian} = 0$, where $D = 0$ indicates phylogenetic conservation and $D = 1$ indicates random trait dispersion in the tree), while the second-strongest signal was for PET ($D = 0.62$; $P_{Brownian} = 0.004$). However, traits for the degradation of most plastics were more randomly distributed around the tree, for instance, PHB ($D = 0.70$; $P_{Brownian} = 0$), polycaprolactone (PCL) ($D = 0.85$; $P_{Brownian} = 0$), and PE ($D = 0.74$; $P_{Brownian} = 0$). Thus, PLA degradation traits appear more conserved and are more likely to originate from a common ancestor than traits for the degradation of other plastics, which may have been more recently acquired.

The structure of a polymer may be defined in terms of its crystallinity. Crystalline polymers usually have very ordered structures, which give them rigidity and strength. In contrast, amorphous polymers generally have more random molecular structures, which allow the polymer chains to move across each other and, consequently, facilitate

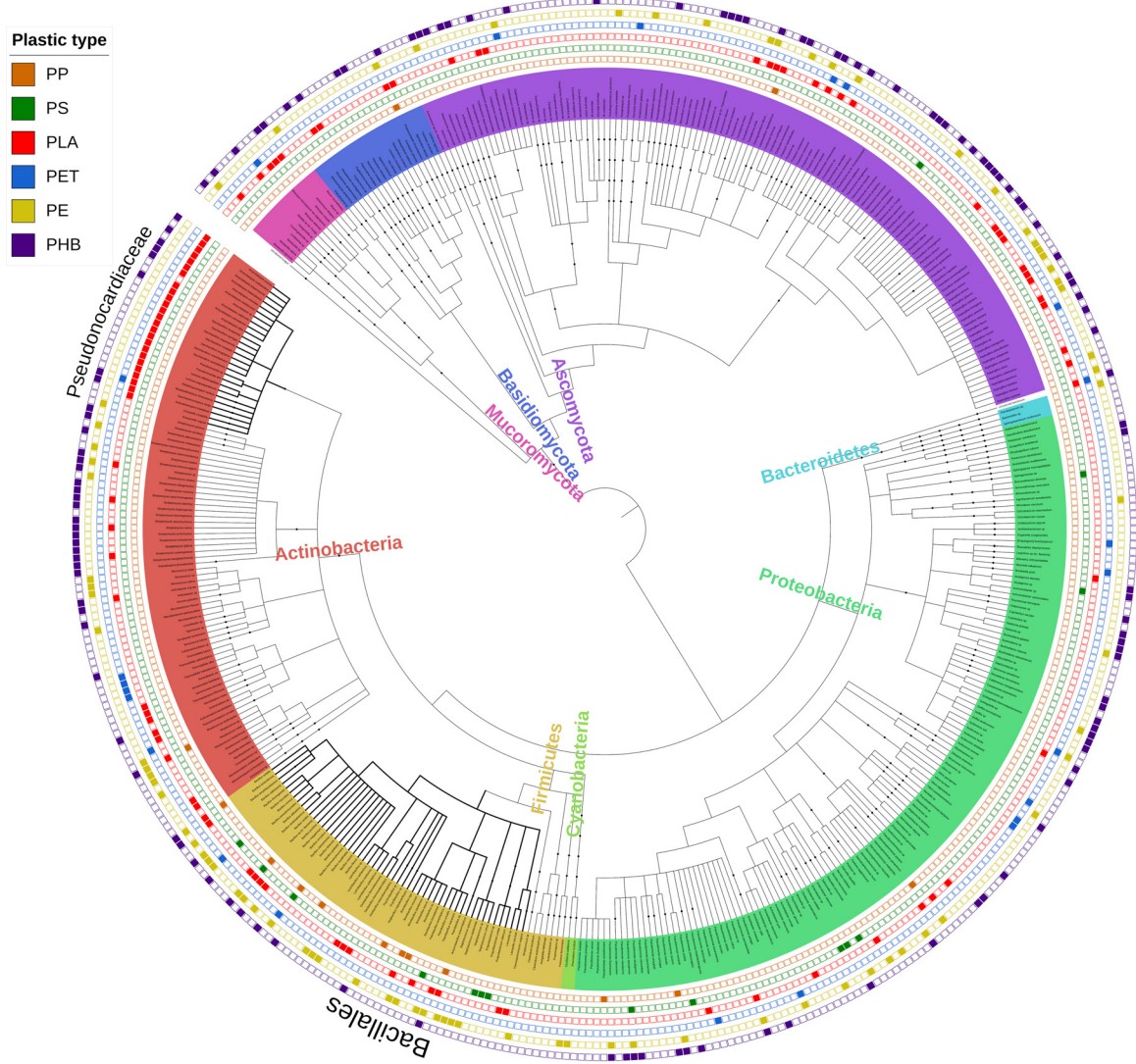

**FIG 1** Phylogenetic tree showing all microorganisms identified as having potential plastic-degrading capabilities. The phylogenetic relationship among species was downloaded from the NCBI taxonomy database. Leaves are colored according to their corresponding phyla. Data points plotted outside the tree represent the ability of each microorganism to degrade each of the plastics shown in the key. The key order from top to bottom is the same as the order of rings external to the tree, from the inside to the outside. Bacteria belonging to the family *Pseudonocardiaceae* and the order *Bacillales* are identified by thicker branches on the tree. The phylogenetic relationship among the reported degraders was extracted from the NCBI taxonomic database classification system. An interactive version of the tree with all plastic types is available at http://itol.embl.de/shared/gambarini with the code P1. All trees used in this publication are each accessible via this website, from P1 to P5; chronologically updated trees are also available and are termed U1 to U3.

the access of enzymes to break down the chains. Microbial degradation of highly amorphous PET films has been demonstrated by Yoshida et al. (26) to be mediated by the PETase enzyme from the bacterium *Ideonella sakaiensis*. However, most PET present in the environment, such as food containers, is manufactured from the crystalline form of the polymer, which the wild-type PETase of *I. sakaiensis* appears to have little activity toward (62). Austin et al. (63) demonstrated that the mutation of two active-site residues improved the degradation of more crystalline PET, highlighting that natural PETase may not be fully optimized for the degradation of crystalline PET. In fact, most studies on microbial plastic degradation do not use the more crystalline forms of the plastics; they instead use aqueous dispersions and emulsions (64), UV-treated plastics (65), or modified polymer films (66). Such challenges indicate that microorganisms lack the ability to efficiently degrade many plastic types in their most commonly

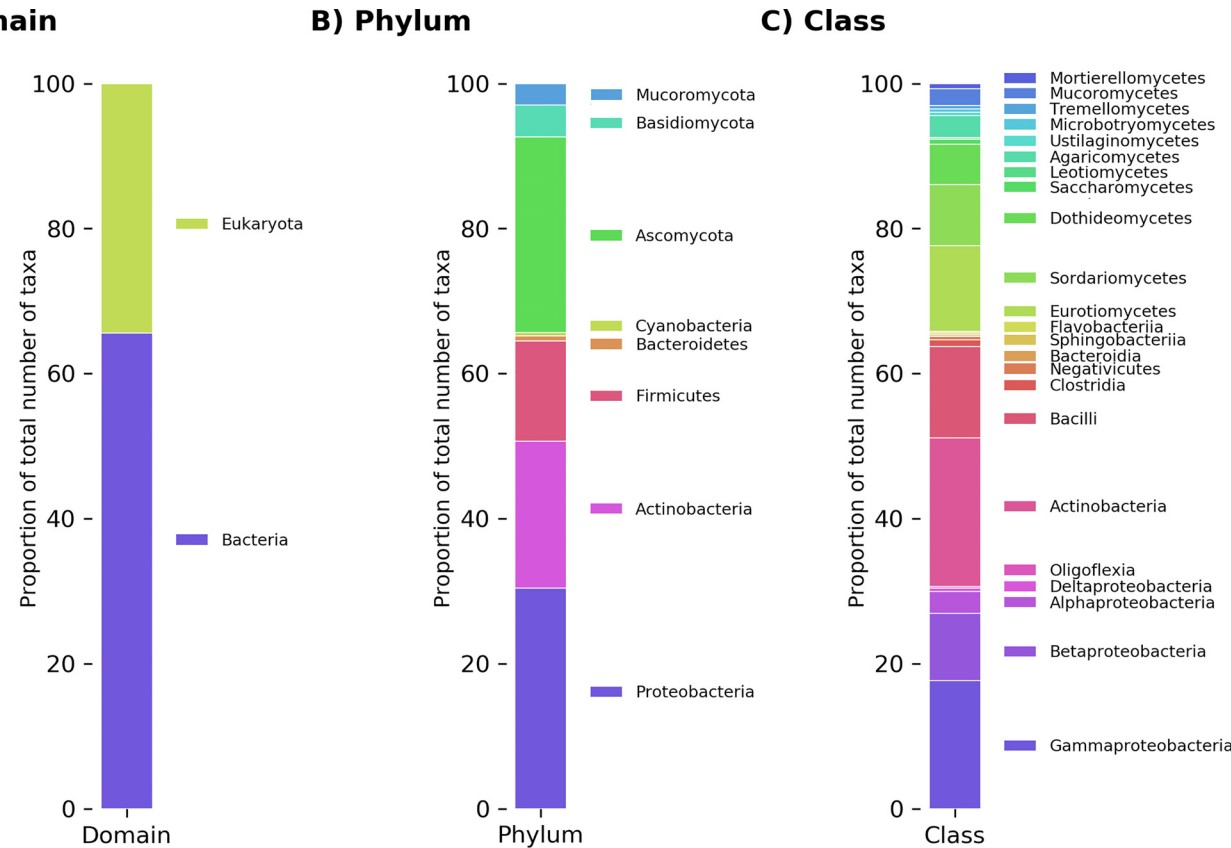

**FIG 2** Relative abundances of all taxa reported to degrade plastics at the levels of domain (A), phylum (B), and class (C).

utilized form (33), but the potential nevertheless exists for the manipulation of microbial genes and enzymes to enhance plastic substrate specificity and rates of degradation, in all forms manufactured.

Most species reported to have plastic-degrading capabilities are bacteria (286 species, or 65.6% of the total number of all species identified) (Fig. 2). Bacterial degraders belong to just 5 phyla from a total of 31 in the NCBI taxonomy database: *Proteobacteria* (30.4%), *Actinobacteria* (20.3%), *Firmicutes* (13.8%), *Bacteroidetes* (0.69%), and *Cyanobacteria* (0.46%). Fungal plastic degraders are represented by 150 species (34.4% of all species identified) within 3 out of 11 fungal phyla in the NCBI taxonomy database. The fungal phyla identified were Ascomycota (27.0%), Basidiomycota (4.4%), and Mucoromycota (3.0%).

Our analysis shows that within this data set, the phylum *Proteobacteria* was the most frequently observed bacterial phylum of plastic degraders, consisting of *Gammaproteobacteria* (17.7%), *Betaproteobacteria* (9.3%), and *Alphaproteobacteria* (3.0%). The genus represented most commonly was *Pseudomonas* (6.7%) (Fig. S5), which belongs to the class *Gammaproteobacteria*. Gammaproteobacterial species were reported to degrade 43 different types of plastic out of the 66 reported in all publications, including most of the highly mass-produced synthetic polymers, such as PE, PET, PP, PS, PVC, and polyurethane (PU). While the degradation of polymers such as PP, PS, and PVC remains controversial (33), gammaproteobacterial species are reported to degrade many plastics that are widely marketed as being biodegradable, i.e., fully mineralizable, resulting in the production of $CO_2$, water, and biomass, including PHB (67), PLA (68), and PCL (69). *Pseudomonas* spp. alone were reported to degrade 35 different plastic types, including most of the mass-produced polymers. Pseudomonads are very adaptable, versatile, and ubiquitous in the environment; these data underline the likely importance of this taxonomic group for the degradation of plastics.

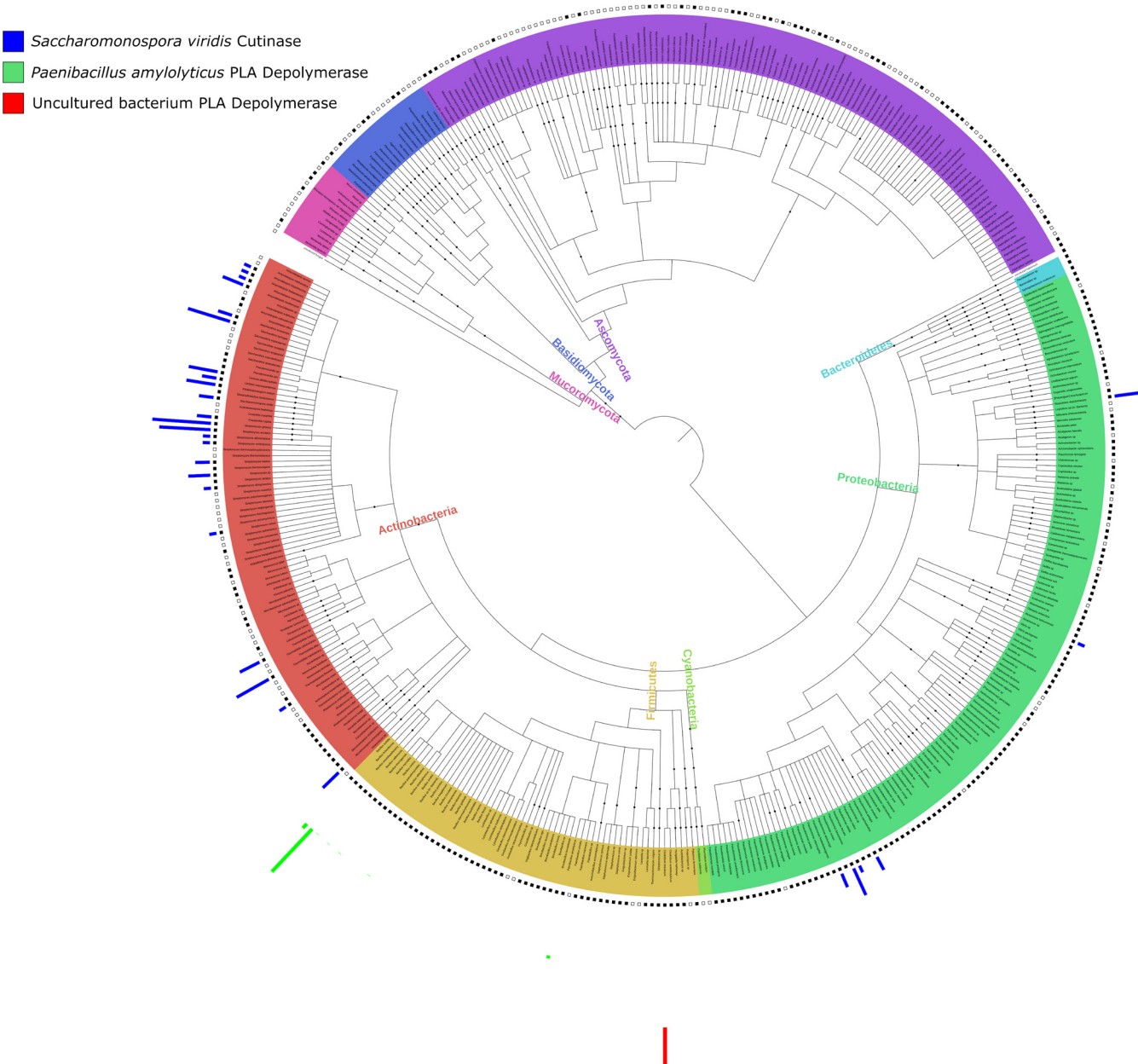

**FIG 3** Phylogenetic tree showing the abundances of genes similar to those encoding *Saccharomonospora viridis* cutinase and *Paenibacillus amylolyticus* and uncultured bacterium PLA depolymerases in all microorganisms with reported plastic-degrading capabilities. Leaves are colored according to phylum. The solid black squares plotted directly around the outside of the tree indicate the availability of genomes in the NCBI genome database. Green bars represent the number of genes found within a genome that are similar to the gene encoding *Paenibacillus amylolyticus* PLA depolymerase, blue bars represent those with genes encoding products similar to *Saccharomonospora viridis* cutinase that has activity against PLA, and yellow bars represent genes similar to a PLA depolymerase from an uncultured bacterium. The key order from top to bottom is the same as the order of rings external to the tree, from the inside to the outside. An interactive version of the tree, with all putative genes related to plastic degradation found by this study, is available at http://itol.embl.de/shared/gambarini with the code P2.

Among the genomes of putative plastic-degrading organisms identified by our search terms, we were able to identify 110 genes corresponding to plastic-degrading activity in 49 different publications. After downloading all available genomes for species with reports of plastic degradation from the NCBI, we identified genes with sequence similarities to those encoding plastic-degrading enzymes within the downloaded genomes. Of all 436 plastic-degrading species in the phylogenetic tree, 281 had genomes for one or more strains available via the NCBI. There are nine genes related to PLA degradation that have been reported to date. PP and PS have no

known genes related to their biodegradation reported. Figure 3 shows the distribution of sequence similarity searches for three of those genes related to PLA degradation. These genes were sequenced from the genomes of *Saccharomonospora viridis* and *Paenibacillus amylolyticus* and also from an uncultured bacterium through metagenomics; all other putative plastic degradation genes can be visualized using the P2 online version of the phylogenetic tree (http://itol.embl.de/shared/gambarini). The *Pseudonocardiaceae* family that has been highlighted here as a source of PLA degraders has 27 species with reports of plastic degradation in our tree. Of these 27 species, 17 species have publicly available genomes, and of these, 11 have genes similar to the cutinase from *S. viridis*. The *S. viridis* cutinase has already been cloned as PET hydrolase, and it has a wide substrate specificity on PLA and other polyesters. The genes and enzymes being used by members of the *Pseudonocardiaceae* to degrade PLA are not yet described in the literature, but these results indicate that a cutinase-like enzyme may be responsible for this activity.

As shown in Fig. 3, the phylum *Actinobacteria* includes multiple species with genes encoding putative cutinase enzymes, as these genes had similarities to the cutinase-encoding gene from *Saccharomonospora viridis* that degrades PLA and PET and also the gene that encodes *Ideonella sakaiensis* PETase, a cutinase-like enzyme that degrades PET. A majority ($n = 50$) of the 88 *Actinobacteria* spp. reported to degrade plastics had genomes available in the NCBI database, of which 23 had genes that are similar to the *Ideonella sakaiensis* PETase and *Saccharomonospora viridis* cutinase genes. As such, this *Actinobacteria* group represents a potential source of PET-degrading enzymes.

In addition to the analysis of reported plastic-degrading taxa, we also investigated the phylogenetic distribution of genes that encode enzymes reported to have plastic-degrading capabilities among all genomes available in the NCBI database for bacteria, fungi, and archaea (i.e., see Fig. S7 in the supplemental material). We identified 16,170 putative plastic degradation orthologs in 6,000 different microbial strains. These strains are assigned to 12 different phyla, of which 5 phyla have no species reported to degrade plastics to date. The five phyla with the largest numbers of similarity matches for plastic degradation genes were *Proteobacteria* (57.4%), *Actinobacteria* (28.9%), *Firmicutes* (10.5%), *Ascomycota* (2.1%), and *Spirochaetes* (0.3%). Most genes were more abundant in organisms closely related to the organisms from which the enzymes were identified, indicating a high degree of vertical transmission. However, there were exceptions. For example, there were 94 genes similar to the PETase-encoding gene from *Ideonella sakaiensis* within its own phylum, the *Proteobacteria*, while 493 similar genes were found among species of the phylum *Actinobacteria*. In addition, the gene that encodes *Ideonella sakaiensis* PETase had three matches to DNA sequences reported as being present within the *Deinococcus-Thermus* phylum, while the gene that encodes *Pseudomonas alcaligenes* polyhydroxyalkanoate depolymerase had 11 hits to members of the phylum *Spirochaetes*. Both these phyla, *Deinococcus-Thermus* and *Spirochaetes*, have no species as yet reported to have plastic-degrading capabilities, highlighting the need to explore other taxonomic groups for plastic-degrading genes and enzymes.

Of the 16,170 putative plastic degradation orthologs identified, the three plastics with the most hits were PHB (10,969), PET (8,233), and PCL (6,809) (Fig. S8). The larger number of PHB-degrading orthologs may be explained by the fact that it is a natural polymer produced by bacteria to store energy. The enzymatic machinery necessary for PHB degradation is likely the product of extensive evolution, facilitating the enzyme's spread among diverse phylogenetic groups. On the other hand, the number of putative degradation orthologs for the synthetic polymers PET and PCL likely reflects the fact that studies of PET degradation are perhaps the most comprehensive for any plastic studied so far (26, 29) and the relative ease of PCL degradation by microbial esterases (70, 71). Other major synthetic polymers such as PP, PS, and PVC still do not have any enzymes reported in the literature to date; therefore, it was not possible to identify any orthologs for these plastic types.

Other probable sources of plastic-degrading microbes with enhanced degradation

potential may be found through the exploration of anaerobic and extremophile taxa because plastic biodegradation may be enhanced by higher temperatures (72), and large amounts of plastic waste are currently disposed of in landfills, thereby ending up under anaerobic conditions (73). In our analysis, we found that 10.4% of the species reported to degrade plastic were able to do so under thermophilic conditions (i.e., at temperatures of >50°C) (all records can be found in Table S1 in the supplemental material). Some studies confirm faster plastic degradation at a higher temperature (i.e., >50°C). For instance, Apinya et al. (55) found that the bacterium *Pseudonocardia* sp. strain RM423 was related to greater PLA weight loss at a thermophilic temperature (58°C) than at a mesophilic temperature (30°C), achieving 74.6% weight loss under thermophilic conditions compared to just 0.9% under mesophilic conditions, after 60 days of the experiment. Similarly, Skariyachan et al. (43) studied plastic degradation at temperatures ranging from 5°C to 55°C and found that the highest biodegradation rates occurred at 55°C, going from 3% ± 2% to 75% ± 2% at 5°C and 55°C, respectively, for low-density PE (LDPE) and from 2% ± 3% to 60% ± 3% at 5°C and 55°C, respectively, for high-density PE (HDPE) (mean percent plastic weight loss ± standard error). Other examples of thermophiles reported to degrade plastic include the bacteria *Thermobifida alba* (74) and *Thermobifida fusca* (75). The low thermal stability of PETase (i.e., from *I. sakaiensis*) has limited the enzyme's ability for efficient PET degradation, although the engineering of increasingly thermostable PETases is considered a possible solution (76).

Given the complex nature of synthetic polymers and their additives, their low rate of degradation, and the lack of tools and technology employed in some of the studies that we analyzed (e.g., not verifying the occurrence or degradation of oligomers and monomers by mass spectrometry, nuclear magnetic resonance, or high-performance liquid chromatography), it is likely that some of the taxa included in this review do not degrade plastic polymers. It may be the case that some studies inadvertently reported the degradation of polymer additives rather than the degradation of the plastic polymer itself. However, it is important to note that when considering all plastic types but PVC, additives average just 4.5% of the total weight of plastics, and some plastic types such as films for food packaging may contain no additives (77).

For all degraders reported here, we compiled information to indicate the strength of evidence available that each taxon degrades plastic polymers. This includes, among other information, the grade of plastic studied and the techniques utilized to verify biodegradation capabilities, which can be visualized on data sets in the online version of the P1 tree (http://itol.embl.de/shared/gambarini) (Fig. S6). Approximately 24.5% of the plastics used were of analytical grade (pure plastic), 17.7% were not of analytical grade, and 57.8% could not be assigned. When analyzing the methods used to identify plastic degradation, clear-zone assays, which measure the formation of a halo around isolates cultivated in agar containing emulsified plastic, were the most common technique, being utilized in 56.4% of the reports. The second most used technique was weight loss measurement. Weight loss alone does not provide strong evidence of degradation, although some studies reported losses of more than 90% of the polymer mass (78, 79). In 96.4% of the reports, weight loss measurements were also supplemented with the use of other techniques such as clear-zone assays, scanning electron microscopy (SEM), and Fourier transform infrared spectroscopy (FTIR). SEM micrographs can indicate surface damage in the form of pits, holes, and cracks formed on the polymer surfaces after degradation. Analyzing specific chemical bonds present in the studied polymers, FTIR is used to confirm changes in these chemical structures after degradation. Overall, while a small number of reports included in our analyses may be false positives, we found that the majority of studies implemented robust practices to confirm the degradation of the polymer (but noting current uncertainly regarding the degradation of polymers, including PS, PP, nylon, and PVC [33]). Since our phylogenetic tree is both updatable and searchable, it provides diverse opportunities for the

scientific community to refine the tree based on further analyses, restricted, for example, to only those organisms for which degradation was confirmed by multiple approaches.

The number of publications reporting plastic-degrading enzymes is rapidly increasing, and our updatable online tree (version U1) enables us to incorporate new plastic degraders as soon as the reports are published. It also allows the interactive visualization of all the data compiled at once. Using our tool, it was possible to identify clusters of plastic-degrading organisms found within the *Pseudonocardiaceae* family and the *Bacillales* order; these clusters suggest that conserved genetic traits may be used by these microorganisms for the degradation of plastics. By analyzing the genomes of microorganisms able to degrade plastics from the NCBI database, we were able to identify that some species have additional genes that might be used to degrade other types of plastic, thereby highlighting new targets for the isolation of plastic-degrading enzymes. These interactive phylogenetic trees provide a foundation for developing a better understanding of the origins, evolution, and phylogenetic distribution of plastic degradation traits, which can facilitate future discoveries in this area, including the discovery of new plastic-degrading microorganisms and enzymes from species that remain difficult to culture in the laboratory and genes capable of degrading multiple plastic targets.

## MATERIALS AND METHODS

For the purpose of this study, we adopt the recent European Union directive (80) definition, in which "plastic" means a material consisting of a polymer to which additives or other substances may have been added and which function as a main structural component of the final products, with the exception of natural polymers that have not been chemically modified. We use the term "natural plastics" to refer to those comprised of polymers resulting from a process that has taken place in nature, irrespective of the process with which these polymers have been extracted. To mine the available literature for evidence of the microbial degradation of all types of plastic, we gathered peer-reviewed publications in two ways: (i) acquiring all publications released up to April 2020 through the Web of Science platform with the search terms [plastic* AND *degradation AND (bacter* OR fung* OR archaea*)] and (ii) capturing all other information that we knew to exist, for example, reports that were already summarized by plastic degradation reviews and all microbes reported to biodegrade plastic present in the PMBD database (81). It is important to note that our literature search was performed with the aim of acquiring a broad overview of the types of taxa reported to degrade plastics, although some taxa, plastics, and enzymes will inevitably have been missed by our search terms; similar searches using terms, including *eukaryot* and diatom*, yielded no results.

The data found in this literature search were compiled in a spreadsheet (see Table S1 in the supplemental material), where for each microorganism reported as being capable of plastic degradation, we recorded the following data, where available: the scientific name, the NCBI taxonomy identification (Tax ID), the type of plastic degraded, the enzyme responsible for degradation, and the sequence of the gene that codes for the enzyme. Additives are blended into many commercial plastics, and although their average content is only a few percent, where used, it can be difficult or impossible to distinguish the degradation of the polymer from the degradation or loss of additives and fillers. We therefore included details regarding whether the plastic used was of analytical or commercial grade such that analyses could also be restricted to additive-free plastic material. The type of evidence for plastic biodegradation occurring was also considered, including observations of the formation of clear zones in plastic-supplemented media, plastic weight loss measurements, and confirmation by nuclear magnetic resonance and scanning electron microscopy. For this reason, taxa putatively identified as being capable of plastic degradation from metagenomics data alone were excluded from our study. The mere isolation of bacteria or fungi from plastic surfaces, plastic-contaminated environments, or medium was not considered evidence for degradation. Table S1 presents details about the analysis techniques, plastic forms, and plastic suppliers used in all studies included in this work.

To ensure research reproducibility and facilitate future updates, a Jupyter notebook (v 6.0.0) (82) (Text S1) was created, containing all of the Python code necessary to analyze these data, build the phylogenetic tree, create the Interactive Tree of Life (83) data sets, and generate all figures for this paper. Updated versions of the phylogenetic trees are released bimonthly to keep up with new reports.

The analysis pipeline started by importing the data spreadsheet (Table S1) into a pandas data frame (84). The NCBI Tax ID of each organism and the corresponding tree topology from the NCBI were acquired through the ETE Toolkit Python API (v 3.1.1) (85). The generated tree and data sets were uploaded to the Web-based tool iTOL (v 4.3.3) (83). Of all 436 plastic-degrading species in the phylogenetic tree, 217 had genome sequences from one or more strains available in the NCBI database. The genomes were first downloaded with the NCBI genome downloading scripts (available at https://github .com/kblin/ncbi-genome-download), and the protein sequences of all plastic-degrading enzymes found in the literature were searched within the genomes using the tblastn algorithm (86) with E value and identity cutoffs of 1e−10 and 50%, respectively. Finally, all fungal and bacterial species Tax IDs for the reported degraders were downloaded from the NCBI. A Python function, random.sample(), was used to

randomly subsample 7,000 Tax IDs from a list object containing all Tax IDs for *Bacteria*, Fungi, and *Archaea* present in the NCBI taxonomy database for comparison to our list of putative plastic-degrading species. Together, the two lists of Tax IDs were used to create a new phylogenetic tree on the iTOL Web server where the comparative phylogenetic distributions of reported plastic-degrading taxa could be assessed.

The phylogenetic distribution of organisms reported to degrade each plastic was assessed by calculating *D* statistics (60) implemented using the phylo.d function in the caper package (v 1.0.1) (61) within the R statistical programming language (v 3.5.3) (87). This approach evaluates if traits associated with a phylogeny are overdispersed ($D > 1$), randomly distributed ($D = 1$), consistent with a model of Brownian motion ($D = 0$), or highly conserved ($D < 0$). To generate a phylogenetic tree compatible with the caper package, we downloaded full 16S and 18S rRNA sequences for all available plastic-degrading species from the SILVA database (88) version 138. For each species found in our research that had sequences available in the SILVA database, one sequence was randomly picked from the database. The resulting sequences were aligned using MAFFT (v 7.429) (89) with default settings, the alignment was trimmed using trimAl (v 1.2rev59) (90) with -gt 0.3 and -st 0.001 parameters, and the tree was built with FastTree (v 2.1.10) (91) with the -gtr and -nt options.

**Data availability.** All data associated with this publication are available in the supplemental material, which contains a spreadsheet with the literature review data (Table S1) and a Jupyter notebook (Text S1) with all Python code used to generate the phylogenetic tree, analyze the data, and create all the figures used in this publication.

## SUPPLEMENTAL MATERIAL

Supplemental material is available online only.

**TEXT S1**, PDF file, 2 MB.
**FIG S1**, TIF file, 0.1 MB.
**FIG S2**, TIF file, 2.4 MB.
**FIG S3**, TIF file, 2.8 MB.
**FIG S4**, TIF file, 1.4 MB.
**FIG S5**, TIF file, 0.3 MB.
**FIG S6**, TIF file, 0.1 MB.
**FIG S7**, TIF file, 1.1 MB.
**FIG S8**, TIF file, 0.2 MB.
**TABLE S1**, XLSX file, 0.2 MB.

## ACKNOWLEDGMENTS

V.G. is supported by a Ph.D. stipend from the George Mason Centre for the Natural Environment (New Zealand). Additional support was provided by the Aotearoa Impacts and Mitigation of Microplastics (AIM[2]) project (Ministry of Business, Innovation, and Employment, New Zealand, Endeavour Fund C03X1802).

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
