## [Reviewer comments · mSystems]

Phylogenetic distribution of plastic-degrading microorganisms

Victor Gambarini, Olga Pantos, Joanne Kingsbury, Louise Weaver, Kim Handley, and Gavin Lear

Corresponding Author(s): Gavin Lear, University of Auckland

Review Timeline:

Submission Date:	October 29, 2020
Editorial Decision:	December 16, 2020
Revision Received:	December 21, 2020
Accepted:	January 4, 2021

Editor: Angela Kent

Reviewer(s): Disclosure of reviewer identity is with reference to reviewer comments included in decision letter(s). The following individuals involved in review of your submission have agreed to reveal their identity: Vinay Pathak (Reviewer #2)

Transaction Report:

DOI: <https://doi.org/10.1128/mSystems.01112-20>

Reviewer #1 (Comments for the Author):**Reviewer****Authors**

The manuscript tries to give a phylogenetic overview on plastic-degrading organisms. The overall topic is timely and certainly interesting. While the manuscript is well written it has some major weaknesses:

A major problem of this manuscript is that the authors do not differentiate between synthetic polymers and natural polymers and most important that they build parts of their story on bacteria and fungi that are only assumed to degrade synthetic polymers. For some of the polymers no enzymes that are involved in the degradation of the polymer are described in the literature and yet the authors have included these in their manuscript. Therefore I would argue that much of the story told here is speculative and aims at a characterization of bacteria and fungi active on additives and solubilizers.

Certainly, there are many different ways in which researchers have sought to confirm plastic degradation, each of which is listed in our supplementary material (Supplementary file 1) for every publication. Details of our criteria are described in our methods section. We concede that there is no single best approach to confirm plastic polymer degradation, but with our updatable and interactive database and tree, readers are provided the opportunity to also conduct their own analyses, selecting criteria that fit their needs (e.g. restricting analyses to only studies conducted on certain polymers, or using specific degradation assays).

Now we make it clear which polymers are natural and which polymers are synthetic. Where appropriate, we discuss these two types of plastics separately. We also added Fig. S9, which identifies the number of orthologs per plastic type incorporating different colors for natural and synthetic polymers.

We only include enzymes that are described in the literature. All the references are listed in our supplementary material and can be further verified. If the reviewer is specifically talking about enzymes for degradation of PP, PS, and PVC, it is true that there are no enzymes described in the literature. Hence, there are no enzymes associated with degradation of these polymers in our database.

The degradation of the polymer versus the degradation of plastic additives and fillers is certainly of substantial research interest. We address this specifically in our text. Further, while we feel confident that we have kept such instances to a minimum, we also make it clear in our manuscript that it is "likely that some of the taxa included in this review do not degrade plastic polymers".

We collated data on the strength of evidence presented for each taxon to degrade each plastic polymer. For example, we verified if the plastics used in every single study were of pure analytical grade, commercial grade (which may therefore contain additives), or where insufficient detail is provided to assess the presence of plastic additives and fillers. We are therefore able to confirm that the key trends described in our data persist even when restricting our analysis to the confirmed degraders of analytical grade plastics. This was not unexpected as, when considering all commercial and industrial plastic types but PVC, additives

	average just 4.5% of the total weight of plastics and some plastic types such as films for food packaging may contain no additives. As already described, since our phylogenetic tree and databases are fully open access and interactive, readers of our article can assess these trends for themselves, using whatever combination of studies they wish, including selecting studies based on the purity of plastic studied. We share the reviewers' concerns that for some plastics more than others (e.g. PVC, PP, PS), uncertainty remains regarding the extent of plastic polymer degradation, if at all. We make clear statements regarding areas of uncertainty in our text, while also referring to a review by Danso et al (2019) relating to this topic.
Further the term plastic-degrading should be defined.	The criteria used to assess plastic degradation potential is already described in our methods, and for every publication, the methods used by the authors are listed in Table S1. (Lines 397-404)
Line 20, to the best of my knowledge there is only a rather small number of bacterial genera known to actively degrade some of the major synthetic polymers such as PU and PET. There is very little - if not any proof that any of the other main synthetic polymers is truly degraded by bacteria.	Yes, several enzymes have been reported as able to degrade PU and PET. For the other major synthetic polymers such as PP, PS, and PVC no enzyme has been described yet – for this reason, no such enzymes are listed in our database. Regarding microorganisms with the potential to degrade these plastics, some have been reported and we include them in our analysis. We, therefore, collated data about the strength of each report found in the literature and we make it clear that for these three synthetic polymers, more evidence is still needed to be 100% sure about their microbial degradation. (Lines 22-25, 149-154, and 299-301)
Line 24, with respect it is not surprising that PLA as a natural polymer is degradable by many different bacteria.	PLA is not a natural polymer. Although the monomer (lactic acid) occurs in nature, the polymer is chemically synthesized and is not produced by any form of life known to date. The fact that PLA has a larger number of microbial degraders reported in the literature is probably linked to its similarity to another natural polymer, a fact already discussed in our manuscript. (Lines 158-162)
Line 36; why not mention that a similar data base already exists already;	We do already mention this in our methods section (now line 385). The fact that other databases do exist is not appropriate for inclusion in our 'Importance Statement'. (Lines 383-384)
Line 40. For the degradation of plastics (i.e. synthetic polymers) presumably different enzymatic activates will be needed. Thus the sentence that more than 14,000 active or	We have now broken the number down by polymer type. We now discuss this in both our abstract and the main text. Further, we have included an extra figure (Fig. S9) to show readers the number of putative orthologs found for each

putative enzymes has been found is relatively weak and should be specified for the different types of plastics.	one of the polymers analyzed in this review. (Lines 291-301)
Line 52, PLA is not part of the 335 mio tons indicated here; the statistics given is not correct.	In terms of global plastic waste, this figure is quite broadly cited. Also adding all global bioplastics production (i.e., not just PLA bioplastics) would have no meaningful impact on this figure, being 'only' in the region of 2 million tonnes.
The introduction lacks a clear definition for plastics and it does by far not cover the current literature on truly plastic-active enzymes.	We now provide a clear and well-used definition of 'plastics' at the beginning of our methods section. As described by our title, our focus is largely on the 'Phylogenetic distribution of plastic-degrading organisms". While we cover the topic of plastic-active enzymes in our paper, this is never stated as a key aim of our study; details of enzymes therefore feature little in our title, abstract, or introduction. In terms of 'current literature' - we have covered all the literature we could capture with our search terms and, also, all literature collated by other reviews. New reports are always being published and the reports released after the date we generated our analysis will be included in the updatable online database. This fact only reassures the importance of our work for the field and makes it clear that an updatable database is highly needed. (Lines 372-378)
Line 104, what were the criteria to state that the polymer was degraded and not the additives. In my view - and this reviewer works in the field- a much smaller number is realistic. In many of the studies mentioned assumed degradation was measured by weight loss or by looking on the surface for alterations and using EM or SEM. It is more likely that in the majority of these studies the additives have been degraded and not the polymer itself. Additives can make up to 50 % (in some cases even 60%) of the synthetic polymers and as correctly indicated in line 117.	When analyzing the methods used to identify plastic degradation, clear zone assays, which measure the formation of a halo around isolates cultivated in agar containing emulsified plastic, were the most common technique, being utilized in 56.4% of the reports. The second most used technique was weight loss. Weight loss alone does not provide strong evidence of degradation, although some studies reported losses of more than 90% of the polymer mass. In 96.4% of the reports, weight loss measurements were supplemented with the use of other techniques, predominantly clear zone assays, scanning electron microscopy (SEM), and Fourier-Transform Infrared Spectroscopy (FTIR). The variety of tools used to confirm plastic degradation in each study is clearly presented and discussed in our manuscript and, since our phylogenetic tree and databases are open access and interactive, readers of our article are free to repeat any analysis, using whatever combination of studies they wish to choose, including by methods used to confirm polymer degradation. In addition, we highlight key common plastics (e.g. PP, PS, and PVC) where, despite some evidence of degradation being reported, not enough evidence yet exists

	to be sure the degradation of pure, high molecular weight polymer has taken place. (Lines 322-354)
Since up to date no truly active enzymes have been identified for the degradation of PP and PS polymers it is very difficult to believe that the reports indicated here have truly identified PP or PS degraders. In fact, it is not even clear which types of enzymes are presumably acting on the C-C bonds of the polymers.	We are unsure how to respond to this statement since we already appear to be in agreement. We found no report of enzymes acting on these two polymers and we are not reporting such. We already state that: "Reports of the degradation of PS and PP must be treated with caution however, since evidence for the degradation of higher molecular weight polymers remains limited, as it does for other polyamide and PVC polymers(30). In most studies, further evidence is required to confirm degradation of the polymer, rather than residual biodegradable monomers such as styrene(55), or plastic additives which comprise a substantial fraction of some plastics, including PVC." (Lines 22-25, 149-154, and 299-301)
Overall a list of the main polymers looked at in this manuscript and the degrading genes and enzymes addressed would help.	Details of all of the polymers, enzymes, and genes (GenbankIDs) are already provided, along with other details in our Supplementary Table 1. (Lines 436-439)
A simple blast search using a potential PETase is certainly not the best way do this.	Our goal with this analysis was to simply show that gene orthologs to the ones shown to degrade plastic are found in many more organisms and taxonomic groups than we have currently described in the literature. The point here was to show that there is a lot of enzymes and taxa that could be targeted for their potential to degrade diverse plastics. For this purpose, we performed a conservative search looking for a high degree of predicted protein sequence-based similarity. Of course, we could compare active sites and get into protein structures, but it is not an objective of our analysis. (Lines 415-419)

Reviewer #2 (Comments for the Author):

Reviewer	Authors
1. Correct the Title - "An update" not suitable word, replace "plastic degrading" with synthetic polymer degrading	We have removed the words "An updated..." from our title. Plastics is a far more familiar term used to describe synthetic and natural polymers. For this reason, we have decided to retain the term 'plastic' in our title, but as suggested by reviewer one, we now clearly define what we mean by 'plastic' and 'plastic-degrading' in our main text.
2. Rewrite line no. 23-23	It is not clear to the authors why line 23 would require re-writing.
3. Add update data in line 62 after 2010 much reports are there on plastic disposal	We now include reference to Lebreton et al (2017) Nat Comm 8: 15611 (Line 70)

4. Rewrite line no. 132-146, the reports of two different polymer mix together if author find is it significant then write separate para for this portion not mix with the synthetic polymer degradation reports	Thanks for this suggestion, we have restricted our text and now clearly address the degradation potential of both synthetic and natural polymers in two separate paragraphs. (Lines 131-173)
5. Check line no. 276	We have rephrased this sentence to make it clearer we are referring to identification of similar genes sequences among these organisms. (Lines 272-275)
6. As the author collects more than 1200 articles on plastic degradation in the supplementary table, but only 257 of 1204 research articles are related to enzymes and genes responsible for plastic degradation. In recent years many new enzymes and pathways have been discovered that are involved in the biodegradation of synthetic polymers. This strongly suggests to authors, only those data who have genomic or proteomic analysis include. All over your articles are well written. But when this paper followed by the other researcher not to create any confusion for selection of polymer degrading microbes	The reviewer is correct that our study is reliant on data from organisms that have positive taxonomic identity. This is made clear from our title “Phylogenetic distribution of plastic-degrading organisms”. However, we agree that this means our analyses may exclude, for example, enzymes isolated from unknown organisms.

December 16, 2020

Dr. Gavin Lear
University of Auckland
School of Biological Sciences
3a Symonds Street
Auckland 1010
New Zealand

Re: mSystems01112-20 (Phylogenetic distribution of plastic-degrading organisms)

Dear Dr. Gavin Lear:

Below you will find the comments of the reviewers. In general, this was well-received, but the reviewers had a few additional recommendations for you to consider. Please address these comments and submit an updated draft of the manuscript. This will not need to go out for review again. Congratulations, and thank you for submitting an interesting analysis!

To submit your modified manuscript, log onto the eJP submission site at <https://msystems.msubmit.net/cgi-bin/main.plex>. If you cannot remember your password, click the "Can't remember your password?" link and follow the instructions on the screen. Go to Author Tasks and click the appropriate manuscript title to begin the resubmission process. The information that you entered when you first submitted the paper will be displayed. Please update the information as necessary. Provide (1) point-by-point responses to the issues raised by the reviewers as file type "Response to Reviewers," not in your cover letter, and (2) a PDF file that indicates the changes from the original submission (by highlighting or underlining the changes) as file type "Marked Up Manuscript - For Review Only."

Due to the SARS-CoV-2 pandemic, our typical 60 day deadline for revisions will not be applied. I hope that you will be able to submit a revised manuscript soon, but want to reassure you that the journal will be flexible in terms of timing, particularly if experimental revisions are needed. When you are ready to resubmit, please know that our staff and Editors are working remotely and handling submissions without delay. If you do not wish to modify the manuscript and prefer to submit it to another journal, please notify me of your decision immediately so that the manuscript may be formally withdrawn from consideration by mSystems.

Sincerely,

Angela Kent

Editor, mSystems

Journals Department
Reviewer comments:

Reviewer #2 (Comments for the Author):

1. Title should be correct (organisms replace with microorganisms).
2. The degradation of plastics as title of the but the missing data regarding key plastic contaminants such as LDPE and HPDE, updated information should also be added.
3. line 59-60- Updated this data up to 2020.
4. line 68-70- Mentioned detailed report which type and kind of major plastic effecting them and mode of action in living system. Specify most affected genera.

Reviewer #4 (Comments for the Author):

Overall, this is a very useful study especially since most research in plastic degradation revolves around putative degraders. Having a phylogenetic tree targeted at these organisms and that is regularly updated will in fact enhance research in this field. However, a gap observed is that the authors do not include any metagenomic studies (whose sequences are published in repositories) as part of their microorganism data, nor offer any explanation as to why they chose to omit it. Following are comments that related to the absence of this data. The reviewer is of the opinion that inclusion of sequencing data sets from plastic degradation studies will enhance the phylogenetic tree and include many more putative plastic degraders, especially those that might not be culturable in a laboratory setting.

Line 127:

Seems that the authors have focused on only those studies that isolate microorganisms and not those that conduct metagenomic studies to include microbial IDs that are not culturable in the laboratory or cannot be isolated in vitro. Since their database is expected to identify putative plastic degraders, non-inclusion of un-culturable species might create a gap in their database. Later in the manuscript they acknowledge that there are very few polystyrene degraders since that polymer has been shown to biodegrade via mealworms only recently. That might be because the current trend, at least for polystyrene degradation, has been to analyze the entire community for species that might be involved in the process.

Line 148:

Authors have not referred the papers by Brandon et al. (2018) (https://pubs.acs.org/doi/abs/10.1021/acs.est.8b02301?casa_token=_MiGVISZEWAAAAAA%3AyjAz-B32htNGSBILUL-rqjHD0hoZL1F5i4Ci3D6-uVoeis5zL3HQeKJVvhULPhQufJRYmJaDoj4c7oAr&) where they conducted a metagenomic analysis on the gut of mealworms and listed two new species *Citrobacter* and *Kosakonia* sp., both from the Enterobacteriaceae family. In line 119, the authors mention that they included species up until April 2020. So, it is surprising that this particular study has not been included.

Line 184-185:

Authors acknowledge that lack of reports for archaea over bacteria and fungi might be due to challenges in culturing them in a laboratory. However, the same challenge is faced by bacteria and fungi as well - this might be another compelling reason for authors to include metagenomic and other sequencing data from plastic degrading studies so as to increase their coverage.

Line 361:

Author mention having analyzed "genomes of microorganisms", yet do not include sequencing data in their study. The fact that this entire study hinges on the creation of a phylogenetic tree and studying the distribution of plastic degrading organisms using that tree, not acknowledging the availability of sequencing data seems like a glaring gap.

Line 432-433:

Were the plastic degrading species somehow mined from the entire SILVA database before aligning them? Or was the database directly mapped against their list of organisms, thereby obtaining only those sequences related to their organism list? Wording is unclear here.

Please also provide version of SILVA database used (132/138) as well as the percent confidence level used for selecting the representative sequences (90/94/97/99). Even though it is mentioned in the supplementary file, including this in the text will make for easier understanding of the protocol.

Comments and Suggestions for the Authors:

Overall, this is a very useful study especially since most research in plastic degradation revolves around putative degraders. Having a phylogenetic tree targeted at these organisms and that is regularly updated will in fact enhance research in this field. However, a gap observed is that the authors do not include any metagenomic studies as part of their microorganism data nor offer any explanation as to why it was omitted. Following are comments that related to the absence of this data. The reviewer is of the opinion that inclusion of sequencing data sets from plastic degradation studies will enhance the phylogenetic tree and include many more putative plastic degraders, especially those that might not be culturable in a laboratory setting.

Line 127:

Seems that the authors have focused on only those studies that isolate microorganisms and not those that conduct metagenomic studies to include microbial IDs that are not culturable in the laboratory or cannot be isolated in vitro. Since their database is expected to identify putative plastic degraders, non-inclusion of un-culturable species might create a gap in their database. Later in the manuscript they acknowledge that there are very few polystyrene degraders since that polymer has been shown to biodegrade via mealworms only recently. That might be because the current trend, at least for polystyrene degradation, has been to analyze the entire community for species that might be involved in the process.

Line 148:

Authors have not referred the papers by Brandon et al. (2018) (https://pubs.acs.org/doi/abs/10.1021/acs.est.8b02301?casa_token=MiGVISZEWAAAAAA%3AyjAz-B32htNGSBILUL-rqjHD0hoZL1F5i4Ci3D6-uVoeis5zL3HQeKJVvhULPhQufJRYmJaDoj4c7oAr&) where they conducted a metagenomic analysis on the gut of mealworms and listed two new species *Citrobacter* and *Kosakonia* sp., both from the Enterobacteriaceae family. In line 119, the authors mention that they included species up until April 2020. So, it is surprising that this particular study has not been included. In fact, nowhere in the manuscript is a reasoning provided for why the authors chose not to include metagenomic data that is already available in certain repositories.

Line 184-185:

Authors acknowledge that lack of reports for archaea over bacteria and fungi might be due to challenges in culturing them in a laboratory. However, the same challenge is faced by bacteria and fungi as well – this might be another compelling reason for authors to include metagenomic and other sequencing data from plastic degrading studies so as to increase their coverage.

Line 361:

Author mention having analyzed “genomes of microorganisms”, yet do not include sequencing data in their study. The fact that this entire study hinges on the creation of a phylogenetic tree and studying the distribution of plastic degrading organisms using that tree, not acknowledging the availability of sequencing data seems like a glaring gap.

Line 432-433:

Were the plastic degrading species somehow mined from the entire SILVA database before aligning them? Or was the database directly mapped against their list of organisms, thereby obtaining only those sequences related to their organism list? Wording is unclear here.

Please also provide version of SILVA database used (132/138) as well as the percent confidence level used for selecting the representative sequences (90/94/97/99). Even though it is mentioned in the supplementary file, including this in the text will make for easier understanding of the protocol.

Responses to reviewer comments:

Reviewer #2 (Comments for the Author):

Reviewer: 1.Title should be correct (organisms replace with microorganisms).

Authors: changed as suggested

Reviewer: 2. The degradation of plastics as title of the but the missing data regarding key plastic contaminants such as LDPE and HPDE, updated information should also be added.

Authors: We're a little unsure what is being requested here. We make it clear in the first sentence of our methods that we follow the EU directive definition of plastics that considers plastic as both the polymer and additives or other substances added as part of the structure. With these details clearly provided in the methods we don't think it is useful to list different types of plastics, or plastic contaminants in the title itself.

Reviewer: 3.line 59-60- Updated this data up to 2020.

Authors: We now provide a reference for global plastics production up until the end of 2019.

Reviewer: 4.line 68-70- Mentioned detailed report which type and kind of major plastic effecting them and mode of action in living system. Specify most affected genera.

Authors: We now explicitly refer to the dominant types of plastics produced, their 'mode of action' in terms of biological impacts and the genera most impacted by these plastics (or at least most reported to be impacted in the literature). (lines 66-93)

Reviewer #4 (Comments for the Author):

Reviewer: Overall, this is a very useful study especially since most research in plastic degradation revolves around putative degraders. Having a phylogenetic tree targeted at these organisms and that is regularly updated will in fact enhance research in this field. However, a gap observed is that the authors do not include any metagenomic studies (whose sequences are published in repositories) as part of their microorganism data, nor offer any explanation as to why they chose to omit it. Following are comments that related to the absence of this data. The reviewer is of the opinion that inclusion of sequencing data sets from plastic degradation studies will enhance the phylogenetic tree and include many more putative plastic degraders, especially those that might not be culturable in a laboratory setting.

Authors: Thanks for these positive comments. In terms of the 'gap' associated with us not including metagenomics data we only included details of taxa for which degradation is stated to have occurred. Metagenomics data can be used to show that certain microbes, or genes, are present, but it does not inform on whether any of the microbes/which microbes present are playing any role in degradation, or that any biodegradation is occurring. Just because a specific microbe is present doesn't mean that it'll degrade the plastic. We now explicitly state this as a reason for why metagenomics data were excluded from our analyses (L414)

Reviewer: Line 127:

Seems that the authors have focused on only those studies that isolate microorganisms and not those that conduct metagenomic studies to include microbial IDs that are not culturable in the laboratory or cannot be isolated in vitro. Since their database is expected to identify

putative plastic degraders, non-inclusion of un-culturable species might create a gap in their database. Later in the manuscript they acknowledge that there are very few polystyrene degraders since that polymer has been shown to biodegrade via mealworms only recently. That might be because the current trend, at least for polystyrene degradation, has been to analyze the entire community for species that might be involved in the process.

Authors: Yes, this is an interesting topic, but again it is hard to confirm plastic degradation from metagenomics data alone. We now clearly state that our analysis is somewhat biased towards organisms which are easier to culture in the laboratory.

Reviewer: Line 148:

Authors have not referred the papers by Brandon et al. (2018)

([where they conducted a metagenomic analysis on the gut of mealworms and listed two new species *Citrobacter* and *Kosakonia* sp., both from the Enterobacteriaceae family. In line 119, the authors mention that they included species up until April 2020. So, it is surprising that this particular study has not been included.](https://pubs.acs.org/doi/abs/10.1021/acs.est.8b02301?casa_token= MiGVISZEWAaaaaa%3AajAz-B32htNGSBILUL-rqjHD0hoZL1F5i4Ci3D6-uVoeis5zL3HQeKJVvhULPhQufJRYmJaDoj4c7oAr&)

Authors: This is an interesting study, but it only shows bacteria associated with plastic consuming mealworm; plastic degradation assays were not performed to confirm that any of these bacterial species can degrade plastics.

Reviewer: Line 184-185:

Authors acknowledge that lack of reports for archaea over bacteria and fungi might be due to challenges in culturing them in a laboratory. However, the same challenge is faced by bacteria and fungi as well - this might be another compelling reason for authors to include metagenomic and other sequencing data from plastic degrading studies so as to increase their coverage.

Authors: In response to prior comments from this reviewer, we now make it clear that we only consider taxa for which plastic degradation is confirmed, experimentally.

Reviewer: Line 361:

Author mention having analyzed "genomes of microorganisms", yet do not include sequencing data in their study. The fact that this entire study hinges on the creation of a phylogenetic tree and studying the distribution of plastic degrading organisms using that tree, not acknowledging the availability of sequencing data seems like a glaring gap.

Authors: All of these data were downloaded from NCBI (using the NCBI Tax ID's in File S1). This is described in our methods, but since our methods are presented at the end of the manuscript, we now also make this clearer in our results section.

Reviewer: Line 432-433:

Were the plastic degrading species somehow mined from the entire SILVA database before aligning them? Or was the database directly mapped against their list of organisms, thereby obtaining only those sequences related to their organism list? Wording is unclear here.

Please also provide version of SILVA database used (132/138) as well as the percent confidence level used for selecting the representative sequences (90/94/97/99). Even

though it is mentioned in the supplementary file, including this in the text will make for easier understanding of the protocol.

Authors: Thanks for this suggestion. We have improved our wording (line 448-450) and now provide the SILVA database version number in the manuscript. In many cases, the 16S or 18S rRNA gene sequences for the organisms reported to degrade plastics in publications is not provided or even sequenced. For this reason, it is not possible to decide which sequences from SILVA to include to construct our tree based on sequence similarity. Therefore, we randomly selected representative sequences representing each species to be included in our tree. We now describe this clearly in our manuscript methods. Our selection of DNA sequence data in this way will have minimal, if any, impact on the topology of our phylogenetic tree.

January 4, 2021

Dr. Gavin Lear
University of Auckland
School of Biological Sciences
3a Symonds Street
Auckland 1010
New Zealand

Re: mSystems01112-20R1 (Phylogenetic distribution of plastic-degrading microorganisms)

Dear Dr. Gavin Lear:

Congratulations, and Happy New Year! Your manuscript has been accepted, and I am forwarding it to the ASM Journals Department for publication. For your reference, ASM Journals' address is given below. Before it can be scheduled for publication, your manuscript will be checked by the mSystems senior production editor, Ellie Ghatineh, to make sure that all elements meet the technical requirements for publication. She will contact you if anything needs to be revised before copyediting and production can begin. Otherwise, you will be notified when your proofs are ready to be viewed.

I hope this gets 2021 off to a good start for you!
Thank you for submitting your paper to mSystems.

Sincerely,

Angela Kent
Editor, mSystems

Journals Department
File S1: Accept

Fig S4: Accept

Fig S1: Accept

Fig S3: Accept

Fig S6: Accept

Fig S2: Accept

Fig S7: Accept

Fig S8: Accept

Fig S5: Accept

File S2: Accept